# Use of strategies to improve retention in primary care randomised trials: a qualitative study with in-depth interviews

V C Brueton,[1] F Stevenson,[2] C L Vale,[1] S P Stenning,[1] J F Tierney,[1] S Harding,[3] I Nazareth,[2] S Meredith,[1] G Rait[2]

[1]MRC Clinical Trials Unit, UCL, London, UK
[2]PRIMENT Clinical Trials Unit, Research Department of Primary Care and Population Health, UCL Medical School, London, UK
[3]MRC Social & Public Health Sciences Unit, University of Glasgow, Glasgow, UK

Correspondence to
VC Brueton;
v.brueton@ucl.ac.uk

## ABSTRACT

**Objective:** To explore the strategies used to improve retention in primary care randomised trials.

**Design:** Qualitative in-depth interviews and thematic analysis.

**Participants:** 29 UK primary care chief and principal investigators, trial managers and research nurses.

**Methods:** In-depth face-to-face interviews.

**Results:** Primary care researchers use incentive and communication strategies to improve retention in trials, but were unsure of their effect. Small monetary incentives were used to increase response to postal questionnaires. Non-monetary incentives were used although there was scepticism about the impact of these on retention. Nurses routinely used telephone communication to encourage participants to return for trial follow-up. Trial managers used first class post, shorter questionnaires and improved questionnaire designs with the aim of improving questionnaire response. Interviewees thought an open trial design could lead to biased results and were negative about using behavioural strategies to improve retention. There was consensus among the interviewees that effective communication and rapport with participants, participant altruism, respect for participant's time, flexibility of trial personnel and appointment schedules and trial information improve retention. Interviewees noted particular challenges with retention in mental health trials and those involving teenagers.

**Conclusions:** The findings of this qualitative study have allowed us to reflect on research practice around retention and highlight a gap between such practice and current evidence. Interviewees describe acting from experience without evidence from the literature, which supports the use of small monetary incentives to improve the questionnaire response. No such evidence exists for non-monetary incentives or first class post, use of which may need reconsideration. An exploration of barriers and facilitators to retention in other research contexts may be justified.

## Strengths and limitations of this study

- This study is the first to explore the use of retention strategies in UK primary care trials and the factors associated with retention and loss to follow-up.
- Interviews were conducted with researchers experienced in UK primary care trials, many of whom had used strategies to improve retention.
- The thoughts and experiences of trial participants have not been gathered to further explore the barriers and facilitators to retention in primary care trials.

areas and communities can be challenging. Inadequate retention can reduce the power of a trial and introduce bias, particularly if drop-out differs across trial arms. Reasons for loss to follow-up can include a change in the participants' location, withdrawal from treatment and/or loss of commitment to the trial, for example, due to complicated treatment regimens.[1] A Cochrane review of strategies to improve retention in trials demonstrated that adding a monetary incentive and offering higher valued monetary incentives increased postal and electronic questionnaire response.[2] Questionnaire response was also increased by recorded delivery of questionnaires, a 'package' of postal communication strategies known as the total design method (TDM)[3] and an open trial design,[4] although these were based on the results of single trials. The evidence of an effect for shorter questionnaires and questionnaires relevant to the disease/condition was less clear. Also, there was no good evidence that the following strategies improved retention: adding or offering a non-monetary incentive, communication strategies (including use of first class mail), behavioural motivational strategies, new questionnaire formats and case management.

Only 6 of the 38 retention trials included in the Cochrane review were embedded in

## INTRODUCTION

Retention in primary care randomised controlled trials conducted across different disease

UK primary care trials and all targeted response to postal questionnaires to improve retention.[5–8] However, primary care trials also require participants to return to sites for clinical outcome measurement to assess disease progression.[9 10] There could be important factors that contribute to the use of strategies to improve questionnaire response and for participants to return to sites for follow-up in the context of primary care trials.

We conducted a qualitative study with primary care trial personnel to explore the strategies generally used to improve retention in primary care randomised trials and to map these to the results of our Cochrane review.[2]

## METHODS
### Sampling the study population
To explore the use of strategies to improve retention in randomised primary care trials, in-depth face-to-face interviews were conducted with principal/chief investigators (PIs), trial managers (TMs) and research nurses (RNs) (table 1) purposively sampled from a sampling frame of primary care randomised trials published from 2000 to 2010. The trials were identified from either the MRC General Practice Research Framework (GPRF) database of clinical research projects, websites of UK primary care research units or from manual searches of trials published between 2009 and 2010 in *Lancet*, *British Medical Journal*, *Family Practice* and *British Journal of General Practice*. The trial publications identified spanned different disease areas, research units and levels of loss to follow-up and were stratified by levels of loss to follow-up ($\leq 20\%$, $\geq 20\%$) and publication date (2000–2004 and 2005–2010) (table 2). All PIs, TMs and RNs were identified for recruitment from the list of authors associated with each trial in the sampling frame or through records of trial staff associated with MRC GPRF primary care trial publications. In addition, TMs were

**Table 1**  Characteristics of interviewees

| Characteristics | Number of interviewees |
|---|---|
| Male | 10 |
| Female | 19 |
| Role | |
| Research nurse | 9 |
| Principal investigator | 10 |
| Trial manager | 10 |
| Location | |
| London | 8 |
| Midlands | 8 |
| Northeast England | 3 |
| Southwest England | 3 |
| East of England | 5 |
| Scotland | 2 |
| Unit | |
| University/research organisation | 23 |
| GP practice site | 6 |

**Table 2**  Other sample characteristics

| Other characteristics | Number of interviewees |
|---|---|
| Number of interviewees from trials published between 2000 and 2004 | 6 |
| Number of interviewees from trials published between 2005 and 2010 | 23 |
| Number of interviewees from trials conducted through the MRC GPRF | 19 |
| Number of interviewees from trials conducted through other research units | 10 |
| Number of interviewees recruited from trials with loss-to-follow-up levels*: | |
| $\geq 20\%$ | 10 |
| $\leq 20\%$ | 17 |

*Two nurses not linked to a sample frame trial.

identified through the Trial Managers Network.[11] TMs recruited in this way had to be associated with a randomised primary care trial published between 2000 and 2010 which was subsequently entered into the sampling frame.

### Data collection
Potential interviewees were sent an information sheet and recruited via email. Prior to the interview the participants were given study information and asked to sign a consent form. Face-to-face interviews explored everyday retention practices and the factors thought to contribute to retention in primary care trials. If retention strategies found by the Cochrane review[2] (ie, incentives, communication methods, new questionnaire formats, behavioural, case management and methodological strategies) were not mentioned spontaneously by interviewees, they were asked specifically about these (Interview schedule online supplementary appendix 1). Interviewees were also asked specifically about the role of the ethics approval process on the use of monetary incentives to improve retention in primary care trials.

The interviews were conducted by one author (VB) in a place convenient to the interviewee; they were recorded digitally and took approximately 1 h. Transcripts were checked following each interview for flow and usefulness of results and the interview schedule was modified accordingly. Appropriate probes were used to allow participants to expand on issues around the use of strategies to improve retention.

### Data management and analysis
We conducted a thematic content analysis. The textual data were managed using Atlas ti V.6. A transcript review group was formed with four authors (VB, CV, GR and FS) and was heterogeneous in terms of the members' professional background. Each transcript was reviewed independently by at least two group members who documented the emerging major themes which were then

discussed by the group in preplanned meetings. This group met monthly while data were collected. There was a high degree of convergence in the themes identified.

Data were also analysed iteratively so that early results could be incorporated and probed in later interviews to increase the depth of the findings. Labels were agreed and used as broad codes to label textual data associated with major themes. Subcodes were identified, discussed and agreed between two authors (FS and VB) and also used to label textual data. Transcripts were subsequently coded by one author (VB). The first two coded transcripts were checked by another author (FS). A key consideration in coding was to mark data produced spontaneously and those which were specifically asked about, therefore data were coded to take account of the response and the question that prompted that response in order to apply appropriate emphasis on responses. Codes for the six strategies identified by the Cochrane review, that is, 'communication', 'incentives', 'questionnaires', ' methodology', 'case management' and 'behavioural' were used as a priori codes for the later part of the interview transcripts as the participants were asked specifically about the use of Cochrane review strategies.

Textual data relating to retention strategies were subsequently retrieved from a coded database and constantly compared within and across the three groups of trial personnel (ie, PIs, TMs and RNs) to identify the use of each strategy and to document the reported advantages and disadvantages of each. Any factor thought by the interviewees to contribute to either retention or loss to follow-up was also coded and retrieved for analysis.[12] [13] Relevant quotes, representing the interviewee's views, were selected to illustrate the results. Interviews were conducted between August 2010 and May 2011 until no new content or themes emerged.

## RESULTS

Fifty-four trial personnel were identified for recruitment from 37 randomised primary care trials from England and Scotland included in the sampling frame. Eleven of the 54 invitees declined to participate (PIs (n=10), TMs (n=1)). Seven of these PIs recommended another coauthor to invite. Fourteen of the 54 did not respond (PIs (n=10), TMs (n=3), RNs (n=1)). Ten PIs, ten TMs and nine RNs were recruited from 23 of the 37 primary care trials identified.

Interviewees were involved in running randomised primary care trials conducted in nutrition, health promotion, neurology, gynaecology, mental health, musculoskeletal, ear nose and throat (ENT), respiratory, endocrine medicine and minor medical conditions. Collectively their experiences, which were not limited to the trial from which they were sampled, included trial design, implementation, coordination and data collection at general practitioner (GP) practice sites by post, face-to-face and electronic methods. Trial publications reported loss-to-follow-up rates between 5% and 39%.

Seventeen interviewees were from trials with ≤20% loss-to-follow-up reported (see table 2). Different types of follow-ups were used in these trials: face-to-face follow-up at clinics (n=20), postal questionnaires (n=6), follow-up at home (n=2) or self-completion diaries (n=2). Combinations of follow-up types were also used: clinic visits or postal follow-up (n=4), telephone and postal follow-up (n=2), registry data and follow-up at home (n=1).

The PIs were responsible for the design and implementation of trials, including research governance and ethics approval applications. Nine PIs were GPs with academic roles, and one was a non-clinical senior academic. The TMs were based in academic research units and managed trial data collection via telephone, post, email and SMS text messages. TMs also coordinated and monitored data centrally and communicated with trial sites. Six RNs coordinated clinical research at GP practice sites. They collected biomedical and clinical data by telephone and face-to-face interview, and communicated with the coordinating centre and with PI clinicians at sites. Three RNs were based at a primary care research network coordinating centre where they answered clinical data collection queries and monitored data quality.

Results are presented first on the interviewee's experiences of and perspectives on using communication, incentive and new questionnaire strategies, identified by the Cochrane review,[2] to improve primary care trial retention. This is followed by views on other less frequently evaluated strategies identified by the Cochrane review,[2] that is, behavioural, case management and methodological strategies. Factors thought to contribute to retention and loss to follow-up in primary care trials are also presented.

### The use of retention strategies identified by the Cochrane review
#### Use of incentives
##### Monetary incentives
Monetary incentives and offers of monetary incentives were mentioned spontaneously by most PIs and TMs as useful strategies to improve response to postal questionnaires. Over half of the PIs interviewed had used monetary incentives given in either cash or voucher format to increase questionnaire return, although there was uncertainty about their effect. The different types used were high street vouchers and mobile phone vouchers, but vouchers were thought of as a burden to administer for trial management teams. Generic vouchers were thought to be more acceptable to all trial participant groups. It was felt that the incentives should be seen by trial participants as a 'thank you' rather than as payment for participation.

The monetary value of the incentives used was between £5 and £20 and this amount was considered reasonable for adults and children. Larger monetary incentives (>£50) were perceived as coercion or bribery by interviewees across the three groups interviewed.

Offers of entry into a prize draw, although seldom used as a strategy to improve questionnaire response, were thought by some interviewees to be a cost-effective alternative to giving a monetary incentive.

There was uncertainty among TMs about effective ways to administer incentives during follow-up to optimise questionnaire response. Some gave a small incentive at different follow-up time points, for example, £5 at randomisation and £5 at trial completion. Others sent an incentive with the questionnaire or on receipt of a completed questionnaire.

TMs and PIs sampled from trials with ≤20% loss to follow-up appeared to use more strategies to improve retention than those from trials with ≥20% loss to follow-up. For example, those in the ≤20% group gave incentives at different time points to try to keep the participants motivated to return questionnaires. TMs and PIs sampled from trials with ≥20% loss to follow-up seemed more cautious about the use of incentives and some felt the participants could feel coerced if incentives were used to improve follow-up.

### Non-monetary incentives

Non-monetary incentive strategies or gifts, that is, pens, certificates of appreciation, key rings and mugs were sometimes used in an attempt to improve participant recruitment and retention. These were used to market trials, and in the case of pens, to encourage and remind participants to return completed questionnaires. Interviewees thought that these gifts could be perceived by trial participants as patronising, not useful or associated with charity fundraising and also as a waste of public money, especially if the participant had altruistic motivations for participation.

> I think people are very sensitive about money especially in the public sector, …if they see something that could be considered wasteful or extravagant, I don't think it would go down too well. Because…if you get involved in a study that's being funded from public funds and from charity, you know, it's quite an altruistic thing to do, I don't think people are looking to be rewarded for it … then they see you wasting money on pens and mugs, I don't think it would go down very well.

> Trial manager interview 7

Some interviewees thought that the money used to finance such non-monetary/gift incentives could be better used to subsidise follow-up visits for participants. Indeed there was consensus among interviewees about the importance of reimbursing travel and parking expenses associated with follow-up visits. Some interviewees differentiated this from giving incentives to participants. Others recounted cases where participants did not want to be reimbursed particularly if they had altruistic motivations for participation, and others felt that an offer of money to cover expenses could make participants feel valued. There was a general feeling that this varied by participants' socioeconomic circumstances, for example, it was thought that participants from higher socioeconomic groups were less likely to accept a monetary incentive.

> We actually gave them ten pounds for their travel costs, once at the beginning and once at the end… and a lot of them didn't want it, you know, they'd say, oh…And I'd say, well just pop it in your local charity.

> Research nurse interview 2

### Ethical approval for the use of incentives

Interviewees across all groups thought it was important that ethics committees sought justification for the use of incentives in trials. They felt that opinion varied among ethics committees about the use of incentives to improve retention but overall they were now more accepting of the use of small monetary incentives and of reimbursing participants' trial-related expenses. Some PIs said that it was helpful to know who was on the ethics committee and that approval to use incentives is generally granted when a clear justification for use is given. Others thought incentives were likely to be approved if they were used to pay participant travel expenses.

## Use of communication strategies

Communication strategies, for example, contact with participants by telephone, letter and email during follow-up, were spontaneously mentioned and routinely used to improve trial retention. RNs and TMs used these as first-line retention strategies.

### Telephone calls

RNs frequently used telephone calls, to arrange or reschedule trial follow-up appointments or to remind participants to return postal questionnaires. RNs and PIs were conscious not to harass participants with too many telephone reminders. TMs thought reminder calls to participants prior to follow-up appointments might increase follow-up attendance. A telephone reminder to a site to remind the RN that a participant is due for a follow-up appointment was also thought to potentially improve retention. PIs did not contact participants by telephone and one thought that it would be 'daunting' for a participant to receive a call from a PI. Nevertheless, PIs were involved in decisions about when to use telephone contact, especially when they thought that participants were fatigued by providing data in diaries or repeat questionnaires.

### Letters

Letters were used predominantly by TMs to communicate with trial participants, but there was uncertainty about their effect. Electronic coloured signatures, franked institutional logos placed on outward envelopes, different coloured envelopes and first class outward post were used to attract the attention of participants.

Letters were usually signed by TMs rather than PIs, the TMs thought that consistency in the signatory was more important than the status of the signatory for improving questionnaire response. Some TMs sent letters to participants in time for weekend delivery, but were unsure whether this influenced response. PIs thought that letters that were personalised, and the tone of the letter was important for retention or improving questionnaire return, but that the associated large mail-outs were burdensome for trial staff. Greeting cards were used occasionally to maintain communication with participants, however, interviewees were negative about the impact of these on retention and questionnaire response, with the exception of the use of birthday cards in paediatric trials and Christmas cards sent to sites from coordinating centres.

> I'm routinely suggesting that we send out birthday cards to children. We've done Christmas cards to participants on behalf of the trial team but there's no evidence that it works, but I think that it's a gesture and it's in the right direction, it's better than nothing.
>
> Trial manager interview 27

There was uncertainty about the effect on retention/ response of the different types of postal delivery used to send trial materials to participants. First class post was used to send trial materials to participants and second class post for questionnaire return. While some TMs and RNs thought that first class post was a waste of financial resources, other TMs thought first class stamped letters were more likely to be opened by the participants. Recorded delivery was occasionally used to send important study-related materials, or to resend questionnaires to non-responders. Some thought recorded delivery was useful to find out if a participant had changed address, but others thought recorded delivery could be inconvenient, for example, if the participant had to collect trial supplies from a post office.

Some TMs noticed that retention improved when questionnaires were sent before clinic visits as this enabled the participants to complete the questionnaire at home beforehand. A prepaid envelope for return of postal questionnaires was thought to reduce the cost of postage to participants and the inconvenience of having to buy a stamp. To minimise respondent burden, clear instructions on how to return the questionnaire once it is completed was thought by TMs to contribute to improved postal questionnaire response.

### Electronic communication

TMs and RNs used emails to contact participants who preferred this approach. SMS text messaging was used less so but most interviewees thought that it would be useful for communicating with young people. The use of an automated system for texting appointment reminders similar to that used by the NHS for clinic appointments was thought to be a potentially useful way to improve trial retention. However, one RN reported that the reminders sent via the system did not improve clinic attendance.

### Newsletters

Most interviewees thought newsletters were useful for keeping participants and site clinicians informed about trial recruitment, retention and general trial news. Others thought these were less useful and could bias trial results by contaminating and confounding the results of the *treatment as usual* group. The frequency with which newsletters were sent to participants varied from fortnightly to annually, and depended on the condition/disease, intervention and length of follow-up. Some PIs thought that having trial-specific information and answers to frequently asked questions as a resource on a website could benefit retention.

### Use of new questionnaire formats

None of the interviewees spontaneously mentioned modifying a follow-up questionnaire to improve response or retention, although some PIs and TMs had used shorter questionnaires for non-responders. When asked specifically about the impact of questionnaire length on retention, interviewees expressed a preference for shorter questionnaires, which they thought increased response. Interviewees also thought that long questionnaires administered frequently led to 'questionnaire fatigue', and some thought that dividing long questionnaires into several short questionnaires could be useful to reduce the burden of data collection on participants. However, some PIs thought that if the questionnaire content was of interest to the participant, then the length of the questionnaire may not affect response. TMs and PIs thought that the involvement of patient representatives in the design and pre-test of questionnaires could improve retention. Many suggestions were made to improve questionnaire design in order to increase response (figure 1).

How a questionnaire was administered and by whom was also thought to affect questionnaire response and retention. Some PIs and TMs thought that if data were collected at the clinic site by a nurse rather than by post that this could increase response.

### Use of other strategies to improve retention

Other strategies identified by the Cochrane review[2] were not routinely used or mentioned spontaneously by interviewees as ways to improve retention. Opinions were mixed about the usefulness of a behavioural strategy where the participants are given information about goal setting and time management to facilitate successful trial completion; methodological strategies where a blind trial design was compared to an open/unblind trial design; and case management where trial assistants manage participant follow-up, by arranging services to enable participants to keep trial follow-up appointments.

**Figure 1** Questionnaire design features reported by interviewees to improve questionnaire return.

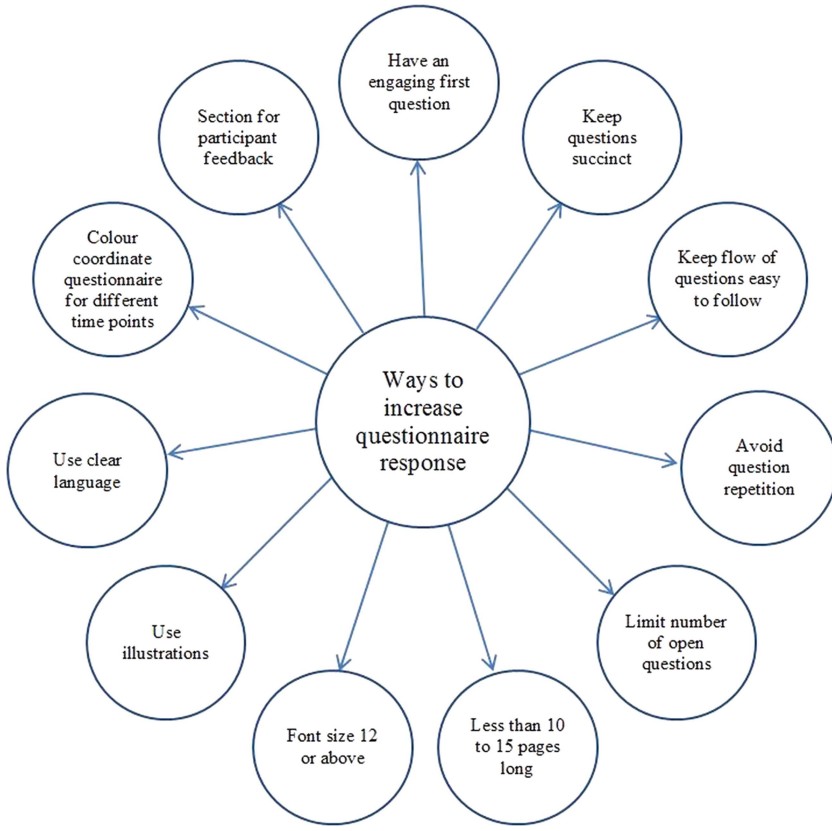

Behavioural strategies were described to interviewees as 'Arranging workshops to give participants information about goal setting and time management,' for example, in the context of an exercise behaviour change intervention. When asked specifically about the use of such a strategy, interviewees were generally negative. Such strategies were discussed in the following ways: 'strange', 'preachy', 'condescending', 'never came across it', 'would never think of doing it', I'd be surprised if it worked' and 'would need evidence that it worked'. Some thought that a behaviour change retention strategy could not be used because it may contaminate a behaviour change trial intervention and therefore may affect the generalisability of trial results.

Although the interviewees were familiar with the concept of blind and open trials, they were not aware of the use of either design as a strategy to improve trial retention. Interviewees felt that using an open trial design to improve retention could bias results, particularly as blinding is used to reduce bias in trials. They also felt that a blind trial design could not be applied to therapist-led behavioural interventions.

One TM had used case management to improve trial retention. This involved the TM arranging services to enable the participants to attend trial follow-up appointments. The interviewees generally thought that case management could improve trial retention with, for example, the elderly, disabled and participants with low incomes but they also thought that case management

could create dependence on the service. The impact of this strategy on retention was thought by some interviewees to depend on the personality of the case manager.

*Factors thought to influence trial retention*
In addition to the strategies employed to improve retention, interviewees also mentioned other factors that had positive and negative influences on retention:

Participants
A key factor thought to optimise retention was to reduce the burden of follow-up for participants in terms of the length of time they spent in follow-up activities and the type of data collected (eg, long questionnaires and bio-medical specimens). More convenient ways to collect outcome data, for example, online data collection were thought to be more effective for retention. Reducing the burden of follow-up visits by streamlining appointments or finding more convenient ways to collect key data (eg, on line or by telephone) were also considered important for retention.

> The burden of activity on the patients, how easy it is for them, how long it's going to take, how intrusive in terms of blood samples or answering huge long questionnaires, it's partly the complexity as well, if it's very simple for them to do, often you just want very simple data back from people, if it can be done either with a, click of a

button by sending a text or an email to people or just … reply paid cards. Do electronic follow-up as much as possible so you don't actually have to bother the patient ….

Principal investigator interview 4

The time participants spend getting to, from and at follow-up visits, giving them adequate time at follow-up visits, and ensuring that they were not kept waiting for follow-up appointments were all considered important factors for retention.

The cost to the patient in time and effort to get to the surgery…what time these appointments would be. How long they would take, if it's a research project, I always feel that patients shouldn't wait, …it's up to the nurse to make … appointments work

Research nurse interview no 1

### Altruism

Many interviewees also perceived that where participants entered a trial for altruistic reasons they were more likely to remain in the trial.

…They [participants] feel as though they want to do good because…often a lot of them say "oh it's payback time", they think the NHS has been good to them so they want to give something back.

Trial manager interview 7

### Perceived benefit of trial participation

Most interviewees thought that if the trial participants perceived an extra benefit from participating in a trial that this would contribute to retention particularly if they were being monitored for disease progression.

In studies where [there] is … constant monitoring … they don't mind giving… samples again and again as part of the trial…But I …think … if they can see some benefit then they'll do it but if they can't or if they've already got that benefit from giving the sample at the first time point they might not want to do it again and again

Trial manager interview 10

Some of the benefits associated with trial participation were extra health monitoring for example: 'ECG' 'BP' and 'cholesterol tests' as well as the extra attention from staff. However, these perceived benefits were thought to be trial dependent.

### Age

Several age groups were highlighted as challenging for trial retention, for example teenagers wishing to opt out of a trial when they reach the age of consent; working mothers juggling school runs with work time and elderly participants who have either lost their independence or

are involved with extended family caring activities. Young men with other interests were also thought to be challenging to retain in trials.

A young girl whose parents had agreed to be in the trial … … she didn't want to and then decided herself she was going to withdraw. And, you know, I can see the potential for that sort of thing if it's a trial in the sort of pre-teen, teen group that,… they might well change their mind…more interesting things going on in their lives, they've got more after school activities or they've got exams or they've got whatever. And coupled with being more independent and wanting to assert their own ideas.

Principal investigator interview 28

In trials with older people, people will die, people will get frailer and maybe be institutionalised and not be able to continue or not want to continue.

Principal investigator interview 28

There are all the usual things, we assume that older people don't move, there's an astonishingly high level of mobility in the older population. So lost to follow-up can occur just because people say "right, I'm going to go and live in…" or "I'm going to go and live near my daughter" or "I'm off to …for the winter". [They] do… a lot of child care, grandparental child care and sorting out their children's divorces and so on and so they get distracted and drop out completely through distraction.

Principal investigator interview 21

### Other participant characteristics

Feelings of guilt and shame with not having achieved positive outcomes in a trial were thought to contribute to loss to follow-up in some disease areas/conditions, for example, in treatment for obesity trials, if participants were not losing weight they might not return to be weighed perhaps due to the shame and stigma associated with not achieving their expected aims. Retention in trials involving behaviour change was also thought to be problematic if change targets were not achieved by participants. For example, interviewees were generally in agreement that retention can be problematic in trials involving participants with a mental illness.

The interviewees felt that participants with chronic diseases may have more interest in the treatment of their condition and return for follow-up visits. Others thought healthy volunteers may drop out of trials because of lack of interest in the trial.

In nutrition studies we're very much bound by looking at healthy individuals to begin with. So from that perspective they may certainly be less motivated than an individual participating in a drug trial who has a severe or debilitating condition that, you know, they want to try and do something about.

Trial manager interview 15

For smoking cessation or weight management or any of these things where a certain behaviour is expected or wanting to be demonstrated. If they're not achieving it they are more likely to just drop out

Research nurse interview 9

Bi-polar groups, a lot of them find it very, very difficult going to an NHS facility for follow-up.

Trial manager interview 17

It depends on [the] area of medicine, for instance, if you're dealing with people with psychiatric illness, there's always a problem with attendance, whether that's within the surgery or whether that's within research.

Research nurse interview 1

### Engaging with participants

Interviewees thought that engaging with the participants during trial recruitment and maintaining a good rapport was key to retention.

It [the recruitment visit] sets the stage for the participants relationship with the trial … it's important to get that right, right from the start… that affects whether people think it's worthwhile coming back…

Trial manager interview 26

At the end of the day it's how the nurse and GP …establish a rapport and a relationship with the participant… that is the key issue in keeping participants in a study.

Research nurse interview 12

I think it is communication at all levels; so the nurse who interfaces with the trial management team at the coordinating centre … listening to the participant's views … and … feeding them back … a two-way process. Because we're this side doing the research, we have to listen … so yes, … communication and understanding [the] population that you're working with.

Trial manager interview 13

Some interviewees across all groups mentioned the impact of trial staff personalities on trial retention. Interviewees thought RNs needed to have a 'flexible approach' and to 'be sympathetic' towards participants, while TMs needed to be 'competent', 'personable', 'persistent', 'enthusiastic' and 'good communicators'.

If you're enthusiastic, can give the information to the participants and…[are] sympathetic to … queries, always being accommodating,[ and] getting back to them if they have any queries …being absolutely spot on … so that they don't feel they've been let down or you haven't been bothered to get back to them.

Research nurse interview 5

I think there's a paradox here because they [trial managers] need to be obsessional about detail but they also need to be relaxed and flexible in their response to different situations ….and problems that come up. That's asking quite a lot, I think. So you know it's great if you find somebody who's good at it…. And prepared to work out of hours!

Principal investigator interview 25

### Tailoring retention strategies

Tailoring specific retention strategies to enable different trial population groups to either return questionnaires or return to sites for follow-up was thought to improve retention. In mental health trials, for example, some interviewees found that participants may wish to be followed-up at home rather than at a clinical site.

Depending on the trial and the length of the trial and your client group …you would have a completely different strategy for… the fifteen to twenty-four year olds, to the seventy-five plus… you do sort of tailor it [the strategy] to your group.

Research nurse interview 2

### Working environment

Some interviewees reported that the working environment at general practice trial sites might impact on retention. It was thought to be important to have a staff member at the site to 'champion' the study. Part-time RNs reported working in isolation and sometimes struggled to find a consulting room for follow-up visits which restricted the availability of flexible follow-up appointment times to offer to participants.

The last study that I worked on…I struggled to find…a room, the practice were incredibly accommodating, and really wanted me to have a room, but it just is very, very tight…it has been, a major issue for me…It was affecting follow up because I was not always able to have a room when I was available, that …would have been potentially available for participants, so it just narrowed down the opportunities for me to offer to participants.

Research nurse interview 5

If somebody genuinely did want to make the trial work then it was possible to make it work…it needed somebody to actually manage the trial within that practice. The practices in which we had success I felt that we had a champion there who was fighting our corner for us

Trial manager interview 27

If somebody comes in and they go to the desk and say, I've come to see X about this trial and, you know, the new receptionist knows nothing. That can be quite off-putting.

Research nurse interview 2

## DISCUSSION
### Statement of principal findings

On the basis of in-depth interviews with primary care trial personnel we identified that incentive, communication and questionnaire strategies are used to try to increase retention in primary care trials. Participant characteristics, for example, interest in the trial, altruism and time, encourage retention. Interviewees thought that participant's perceptions of benefits from participation; the age of participants, for example, young men and teenagers; the disease area/condition, for example, mental illness and behavioural problems; and the research environment in which the trial was conducted, for example, availability of consulting rooms for follow-up appointments all contributed to loss to follow-up in trials.

### Strengths and weaknesses

Research on strategies to improve retention of participants in primary care trials is limited and commonly grouped with recruitment strategies.[14] [15] Trial retention is important because loss to follow-up can lead to incomplete data for the primary outcome, bias results and impact the generalisability of trial findings. This study is the first to explore the use of retention strategies in a wide range of randomised UK primary care trials and to identify factors associated with retention separate from recruitment strategies. Interviews were conducted with researchers experienced in the design, leadership, management and implementation of randomised UK primary care trials, many of whom reported that prior to this study they had not thought deeply about loss to follow-up or strategies used to overcome this. With the focus by funders on recruitment rather than retention targets, this study specifically raises awareness of the complexity of retention in primary care trials and the many strategies and factors used to maximise retention. Although the thoughts and experiences of trial participants were not captured to further explore the barriers and facilitators to retention the results show that trial personnel have an understanding of the challenges participants encounter to stay in trial follow-up. It is unclear how generalisable the results are to randomised trials outside of primary care.

### Meaning and implications for the use of retention strategies

These results enhance the interpretation and provide insight into the implementation of the results of our Cochrane review on ways to improve retention in trials.[2] Together, these two studies inform the future use of effective strategies to improve retention in primary care trials.

There was general agreement that small monetary incentives were likely to be viewed favourably by ethics committees and led to increased questionnaire response, in agreement with the findings of the Cochrane review.[2] Although there was uncertainty among those interviewed about whether monetary incentives should be given up front or offered on receipt of data, the Cochrane review showed that either approach increased the postal questionnaire response.[2]

Scepticism among interviewees about non-monetary incentives and their value was also supported by the Cochrane review which did not find them to be effective. Trialists may therefore need to reconsider the use of this type of incentive.

A range of different communication strategies were discussed by interviewees. Although trialists routinely preferred first class post, and the use of enhanced letters, the Cochrane review found no evidence to support either approach. Therefore, second class post and standard letters can probably be used for postal correspondence with the primary care trial participants, with little impact on loss to follow-up.

Trialists had mixed opinions on the use of recorded delivery, nevertheless recorded delivery of questionnaires compared with a telephone reminder was found to be effective for improving questionnaire response.[2] [5] To avoid any inconvenience associated with the use of this strategy, preplanning the delivery time with the participant with a reminder to expect a recorded delivery of trial materials is recommended if this strategy is to be used in future trials.

Different types of reminders and alternative questionnaires are routinely used by primary care researchers to improve trial follow-up; however, there is no clear evidence that reminders and prompts in addition to standard trial follow-up procedures are beneficial to retention or to improve questionnaire response.[2] Similarly, there is no evidence to suggest better response rates to shorter questionnaires.[2]

Furthermore, interviewees were particularly negative towards the use of behavioural strategies and there was no effect found for these in our Cochrane review.[2] Case management, although seldom used, was thought to be potentially useful by interviewees; however, there is no evidence that this is effective.[2] The interviewees also thought that blinding participants to their allocated intervention would improve retention. This is in contrast to findings from the Cochrane review which found that an open trial design was more effective than a blind design in one trial.[2] [4] Avenell[4] argues that double blind trials do not reflect usual healthcare procedures, and that open trials, where the intervention is compared with no treatment or usual treatment, could give a better measure of the differential effects in normal care settings. The open trial design, if used as a stand-alone strategy to improve retention, would need to be considered in the context of the overall aims and objectives of the trial. Further evaluation is needed in different trial contexts and settings if this retention strategy is to be adopted.

Several other factors and ways to improve retention were identified that were not covered in our Cochrane review.[2] Some of these such as relationships and communication

between trial staff and trial participants are clearly more difficult to design interventions for and to evaluate. The factors identified indicate a need to train and support trial staff in ways to improve trial retention. It seems clear from this qualitative study that trial staff were oriented to participant concerns and needs (as they perceived them) and had a high level of empathy with participants.

### Integrating qualitative research with Cochrane reviews

Integrating the results of this qualitative study and the Cochrane review helped us understand the different experiences primary care researchers have with the strategies identified by our Cochrane review.[2] The qualitative results support some of the results of our Cochrane review and demonstrate the potential transferability of the different effective retention strategies to trials conducted in primary care. For example, incentives were employed in different ways to try to improve responses to postal questionnaires. The results have also helped identify other factors that might promote retention such as building a rapport with trial participants, and potential barriers like finding a consulting room for follow-up appointments. This qualitative approach could be replicated in research contexts apart from primary care to determine how appropriate and useful the strategies identified by the Cochrane review[2] are in practice.

While the absolute effects of effective strategies identified by the Cochrane review are modest,[2] this qualitative study highlights that multiple strategies are often used to try to achieve maximum retention in primary care trials. Furthermore, it suggests that clinical trialists should consider other factors that may affect retention, such as building a rapport with participants and minimising the impact of trial follow-up on their daily lives. Consideration should also be given to the characteristics of the intervention, participants and trial personnel when choosing effective strategies identified by our Cochrane review to improve trial retention.[2]

### Future research

It was clear from the interviews that there is a move towards contacting participants by email and a desire to use text reminders when automated facilities for these become available for use in randomised trials. These new strategies would need evaluation in the future trials.

Further research is also needed to identify potential barriers and facilitators to follow-up in trials, for example, with trial participants from different social, economic, age and disease groups. The results would be useful for primary care researchers and would help target and tailor strategies to meet the needs of different population groups to keep them specifically engaged in trial follow-up.[16]

### CONCLUSION

Qualitative research can help to interpret the results of systematic reviews by telling us what researchers do or think happens in research practice. These findings provide a deeper understanding of the factors that may facilitate retention such as rapport with participants and respect for their time, and highlight the participant and environmental characteristics that can limit the implementation of effective strategies. The use of small monetary incentives to improve questionnaire response is acceptable; however, a reconsideration of the use of non-monetary incentives, and certain communication strategies including first class post is needed. A similar exploration of barriers and facilitators to retention in other research contexts would help identify ways to improve retention in trials conducted in other research contexts.

**Acknowledgements** The authors would like to thank all 29 interviewees who gave their time and thoughts on the complexities of trial retention in primary care.

**Contributors** VCB, FS, GR, SM, SH and IN conceived the study idea. VCB applied for ethics approval and collected the interview data. VCB, GR, FS and CV directed the analyses, which were carried out by VCB. All authors participated in the discussion and interpretation of the results. VCB wrote the initial manuscript draft. All the authors critically revised the manuscript for intellectual content and approved the final version.

**Funding** This project was funded by the Medical Research Council Population Health Sciences Research Network grant number PHSRN 30.

**Competing interests** None.

**Ethics approval** University College London research ethics committee. Application no. 2342/002.

**Provenance and peer review** Not commissioned; externally peer reviewed.

**Data sharing statement** No additional data are available.

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
