## [Reviewer comments · BMJ Open]

Some articles will have been accepted based in part or entirely on reviews undertaken for other BMJ Group journals. These will be reproduced where possible.

ARTICLE DETAILS

TITLE (PROVISIONAL)	Use of strategies to improve retention in primary care randomised trials: a qualitative study with in-depth interviews
AUTHORS	Brueton, Valerie; Stevenson, Fiona; Vale, Claire; Stenning, Sally; Tierney, Jayne; Harding, Seeromanie; Nazareth, Irwin; Meredith, Sarah; Rait, Greta

VERSION 1 - REVIEW

REVIEWER	Pitkethly, Marie University of Dundee, Scottish School of Primary Care
REVIEW RETURNED	18-Sep-2013

GENERAL COMMENTS	1. The Abstract largely reflects the paper, but I would have liked to see "making efficient use of participants' time" included 2. Page 18, line 20/21: I don't agree with the interpretation of Edwards et al's review – he actually found that in the 2 studies that compared telephone and postal reminders, postal was the most effective. The odds of response were almost one third higher when follow up contact (either repeat mailings or telephone calls) was made. 3. Good example of the complementary use of qualitative and quantitative methodology. Views of the trial participants would be invaluable and I would have liked that to be the focus of future research in the conclusions, rather than extending to other research contexts.
---

REVIEWER	Ulucanlar, Zelda University of Bath
REVIEW RETURNED	24-Sep-2013

GENERAL COMMENTS	This paper reports from a qualitative (interview) study of trial staff's experiences with and views on strategies to increase participant retention and questionnaire response in primary care RCTs. While the dataset appears to have yielded some interesting and important insights, the analysis in this paper fails to present these in a conceptually meaningful and coherent manner. It is difficult to judge without having access to the dataset, but the findings read like a descriptive list of views with little indication of an analytic attempt to elicit the more salient findings and their conceptual significance. The objective and research questions seem unclear and there are also problems with the reporting of the methods. In my view, the paper needs to be extensively revised before being considered for publication.
---

Introduction:

The aim of the qualitative study is stated as: to gain insight into how best to improve retention in primary care and RCTs in general. However, in the Discussion, there is reference to a mixed method approach (i.e. the Cochrane review + qualitative study) and to the qualitative study's role in exploring the acceptability and feasibility of the strategies identified in the review. The research aims, objectives and questions need to be made clearer at the outset. Furthermore, the design doesn't seem appropriate if the question was to elicit 'how to best improve retention'; the study is about trial staff's views, not observed practices or evaluation of those practices.

Also, the Introduction could be better organised by moving the first part of second paragraph (p 5, L 23-29) to the beginning of the first paragraph, then introduce the Cochrane review and its limitations.

Methods:

It is not clear how interviewees were identified. While trials were identified from websites and published reports (P5 L 56), trial managers and nurses were identified through networks – why (p 6, L 7)? Were the PIs, trial managers and nurses interviewed affiliated to the trials identified from the literature or with different trials? If the latter, what was the purpose of identifying trials? On the other hand, the authors say that 'potential interviewees were identified from the list of authors...' (P 6, L 11)?

'All transcripts were reviewed and validated by at least two authors' (P 6, L 45). What do the authors mean by 'validating transcripts'?

Was each transcript coded by two authors? If so, what was the degree of convergence/divergence and the procedure used to discuss and resolve differences in coding? Again, how were the codes 'verified' by another author? Did the other author actually code transcripts or simply review the codes independent of transcripts? Details of the mechanics of Atlas use are not needed. I imagine the six strategies were used as a priori codes for the latter part of the interview where the interviewees were asked about these. How were the 'open' sections of the interview coded where interviewees were asked to talk about any strategies they used? Were the six strategies used as codes for these sections as well? The authors describe a process of constant comparison – if this is what they used, it would be good to indicate this.

A positivistic language is used: 'advantages and disadvantages' of each strategy were 'identified.' The authors need to acknowledge that they are reporting views and use more appropriate language, i.e. the reported advantages, etc.

I find the analysis superficial and conceptually weak. However, this may reflect the type and function of the study.

Punctuation needs to be reviewed.

Results:

An introductory paragraph with the main findings and the focus of the analysis would be helpful to orient the reader through the many sub-headings. I've found it hard to anchor the paper; the problem seems to be that it's trying to do two different things: a) investigate the use of and views on the Cochrane strategies and b) explore the everyday experiences and views of trial staff. From my reading, the most striking finding is the difference between the types of intervention studied in the Cochrane review and the issues that preoccupy staff on the ground and the strategies they have developed. I think a paper structured around this core idea would work much better and the authors need to consider if this is feasible. Given the general aim of the study (in the Introduction) and the ordering of the questions, with the Cochrane review strategies at the end of the topic guide, I'm not sure why these are presented as the

main findings, with the 'open' sections of the interview – interviewees' own experiences and views – left to the end. The authors should consider turning this around, particularly as the open section has produced interesting and important data.

The authors note, in the Introduction, the absence in the Cochrane review of strategies used to encourage trial participants' return to study sites for clinical follow-up (as opposed to strategies used to encourage questionnaire response). They seem to imply that this is part of the rationale for the qualitative study which was designed to address this by including interviewees from primary care trials where this type of retention (return to study sites) was a problem. Yet, they give no information on the nature of the follow-up procedures in the 37 trials. As a result, the reader does not know what type of retention problem (return to site/return of questionnaire) the various strategies discussed by the interviewees were intended to address. Having identified this as an important variable, the authors need to provide information on this. Similarly, the authors do not report on whether there were any patterns in the data on the basis of level of loss to follow-up (a sampling criterion).

What the authors have labelled as 'the disease area' (as a factor in retention) may be more appropriately and informatively labelled as 'guilt and shame.'

Box 1 and Box 2, as currently organised, are confusing and they fragment the reader's attention as the main text already has these themes and relevant data extracts. If data extracts for the themes are to be put in a box, then they should all be put in a Box. It's confusing to have extracts in the main text and then more in a Box. Tables 3 & 4 don't work either. Some information is repeated in the main text; the information includes: who used the strategy, whether it was mentioned spontaneously, how it was used and views on effectiveness, but in a haphazard way, i.e. not all these domains are covered in each cell. The result is a confusing collection of disparate information. Table 4, in particular, reads like (under-analysed) descriptive summaries of views and experiences on different aspects, reported in different ways. Tables could be used to supplement (not repeat) the main text and need to provide information in a consistent manner.

Minor points:

P7 L25: Should '20' be '17'

P11 L44-48. What does this sentence mean – which aspect of the questionnaire could influence the response?

P14 L43 – there seems to be text missing here.

Referring to the Cochrane systematic review as the Cochrane review may make the source clearer in each instance

Table 3: Should the distinction between 'pharmaceutical' and 'academic' trials be: 'industry sponsored' and 'publicly funded'?

Table 4: 'Speculation'? Or 'experiences and views'?

Discussion:

The authors need to emphasize the fact that only two (incentives and communication) of the six Cochrane review strategies were mentioned spontaneously. On the other hand, several other factors and ways of improving retention were identified that were not covered in the Cochrane review. Some, e.g. the relationship/communication between trial staff and trial participants, are clearly more difficult to design interventions for and to evaluate. It seems clear that trial staff oriented to patient concerns and needs (as they perceived them) and had a high level of empathy. This seems an important finding with implications for future intervention design as well as the appropriateness of existing ones (behavioural strategies), but is not picked up in the Discussion. The Discussion

	has to be revised if the authors choose to reorganise and refocus the paper as suggested here.
--	--

REVIEWER	Cupples, Margaret Queens University, Belfast, General Practice
REVIEW RETURNED	30-Sep-2013

GENERAL COMMENTS	The research question and aim are clearly defined at the end of the introduction but could be better clarified in the abstract. In respect of the methods, there is inconsistency in description of 'trials' and 'randomised trials' - this should be clarified - that all studies were RCTs? It should also be noted that all nations in the UK were not included- studies selected were confined to England and Scotland. Also, no information is given about those who were selected for interview but declined to participate - this should be included. The description of identification of participants would be better if attention was paid to detail in a logical flow of the process followed was given - perhaps a table could show the sources from which participants recruited -and the relevant trials? Also - how were trials with 20% loss classified - or were they excluded? And what about the publications from 2005? How was consent obtained? The setting of interviews (were they face-to face?) may be added to the section on data collection. I have said that the results are not presented clearly - because I feel they could be presented much more clearly. Does the statement that interviewees were from the 'fields of' - does this mean they had specialist interests in specific areas - or does this refer to the publication from which they were identified, or the studies which they used in their responses? Perhaps a version of Tables 3 and 4 could be included and referred to early in the results, to help to clarify. I found it was confusing to read about types of incentives/ values/ ethics committees' views/ perceptions of use/timing/alternatives/ethics approval - all in one section. More time taken to consider how these findings could be considered to report the 'use of strategies' and them to report 'factors thought to contribute to retention' - ie a clear statement of findings regarding the perceived value of incentives would be helpful. With regard to Tables 3 and 4, it is not clear why they are divided into the 3 groups- nor where the different groups share experiences/ views re strategies. Consideration of a more organised framework for reporting these would be helpful, as would a consistent approach to the detail reported regarding responses. Similar comments re organisation of the report of findings applies to each of the sections - communication/ questionnaire format/ - to make the actual findings more easily identified within the text. In respect of the section re the use of other strategies - I think these needs more detail - I find it difficult to understand how 'Arranging workshops about goal-setting.....' relates to a retention strategy rather than being a component of the intervention being tested?? The opening paragraph of the discussion suggests that age/ condition/ research environment contribute to loss of follow-up - but the specific characteristics of these variables which are relevant need to be identified in order to provide meaningful information. In respect of the discussion of limitations, again I'm not sure why there's value in a study which focuses only on retention rather than recruitment as well? Can this be elucidated? Or perhaps be explicit
---

regarding how this paper adds value to previous work in the UK and how these findings compare with previous reports? The use of the wording 'RCTs in general' may be reviewed, to be more specific about what 'in general' means?

The section headed 'meaning and implications' would be more clearly presented with sub-headings and this may help improve the coherence within sub-sections. (there is a typo - line 4 of page 18 - 'if' - should be 'of'.)

I find the section re integrating qualitative research with systematic reviews rather vague - this could be much more meaningful if specific examples were used to illustrate statements.

Lastly, the focus in the conclusion is on incentives and 1st class post but the previous section highlights the use of text and email and I found comments regarding reminders, harassment, engagement, rapport, patient benefit and respect for patients' time all very useful pointers towards improving retention - could these be integrated in the conclusion?

If you have any further comments on the paper please enter them below.

I am aware that I am being critical of this paper - but I do feel that it has value which could be much improved by added attention to detail and organisation of its presentation.

This is an interesting study but I feel its contribution to the literature would be much improved by some further work. The introduction itself does not read easily - small changes to the use of English would help its fluency and allow better coherence in the development of the rationale for the study. The closing words - 'trials in general' is a rather loose statement and should be better defined. A minor point re grammar - traditionally the correct use of data is plural - thus data 'were' rather than data 'are' would be correct.

VERSION 1 – AUTHOR RESPONSE

Reviewer Name Marie Pitkethly

Institution and Country Scottish Primary Care Research Network, University of Dundee, UK

Please state any competing interests or state 'None declared': None declared

1. The Abstract largely reflects the paper, but I would have liked to see “making efficient use of participants’ time” included

We have amended the abstract to read

There was consensus among the interviewees that effective communication and rapport with participants, participant altruism, respect for participant’s time, flexibility of trial personnel and appointment schedules, and trial information improve retention.

2. Page 18, line 20/21: I don’t agree with the interpretation of Edwards et al’s review – he actually found that in the 2 studies that compared telephone and postal reminders, postal was the most effective. The odds of response were almost one third higher when follow up contact (either repeat mailings or telephone calls) was made.

Edwards found pre-contact by phone vs. mail, first response 1.40 [1.02-1.93] Z = 2.07 (P = 0.039). Final response was OR 1.18 [0.77, 1.80] Z = 0.76 (P = 0.45) page 391 Edwards review (2009). We used final response in our Cochrane review discussion.

3. Good example of the complementary use of qualitative and quantitative methodology. Views of the trial participants would be invaluable and I would have liked that to be the focus of future research in the conclusions, rather than extending to other research contexts.

We agree that the views of the trial participants would be invaluable and we have added a comment about this to the section entitled strengths and weaknesses of the study in the discussion section to read as

A limitation of this study is that the thoughts and experiences of trial participants were not captured to further explore the barriers and facilitators to retention.

We have updated the section on future research to read as

Further research is also needed to identify potential barriers and facilitators to follow-up in trials e.g. with trial participants from different social, economic, age and disease groups. The results would be useful for primary care researchers and would help target and tailor strategies to meet the needs of different population groups to keep them specifically engaged in trial follow-up¹⁷.

Reviewer Name Zelda Ulucanlar

Institution and Country Cardiff University

Please state any competing interests or state 'None declared': none

Introduction:

1. The aim of the qualitative study is stated as: to gain insight into how best to improve retention in primary care and RCTs in general. However, in the Discussion, there is reference to a mixed method approach (i.e. the Cochrane review + qualitative study) and to the qualitative study’s role in exploring the acceptability and feasibility of the strategies identified in the review. The research aims, objectives and questions need to be made clearer at the outset.

We have made the aim of the study clearer, and reflected this in the abstract which now reads

We therefore conducted a qualitative study with primary care trial personnel to explore the strategies generally used to improve retention in primary care randomised trials and to map these to the results of our Cochrane review².

2. Furthermore, the design doesn't seem appropriate if the question was to elicit 'how to best improve retention'; the study is about trial staff's views, not observed practices or evaluation of those practices.

The study was about trial staff views on the use of retention strategies, some of which they would have experience of using, we did not directly observe or evaluate their practice.

3. Also, the Introduction could be better organised by moving the first part of second paragraph (p 5, L 23-29) to the beginning of the first paragraph, then introduce the Cochrane review and its limitations.

We have reorganised the introduction.

Methods:

4. It is not clear how interviewees were identified. While trials were identified from websites and published reports (P5 L 56), trial managers and nurses were identified through networks – why (p 6, L 7)?

Potential interviewees were identified from the list of co-authors of trials listed in the sampling frame. Where we could not identify TM's and RN's we contacted primary care research networks, TM's and RNs identified through this method had to be linked to a published trial (co-author or not) which was included in the sampling frame.

We have changed this in the methods section which reads

To explore the use of strategies to improve retention in primary care trials, in-depth face to face interviews were conducted with principal / chief investigators (PIs), trial managers (TMs) and research nurses (RNs) purposively sampled from a sampling frame of primary care trials published from 2000-2010. The trials were identified from either the MRC General Practice Research Framework (GPRF) database of clinical research projects, websites of UK primary care research units or from hand searches of trials published between 2009 - 2010 in Lancet, British Medical Journal, Family Practice and British Journal of General Practice. The trial publications identified spanned different disease areas, research units, and levels of loss to follow-up (Tables 1 and 2) and were stratified by levels of loss to follow-up (<20%, ≥20%), and publication date (2000-2004 and 2005-2010). All PIs, TMs and RNs were identified for recruitment from the list of authors associated with each trial in the sampling frame or through records of trial staff associated with MRC GPRF primary care trial publications. In addition TMs were identified through the Trial Managers Network¹¹. TMs recruited in this way had to be associated with a primary care trial published between 2000-2010 which was subsequently entered into the sampling frame.

5. Were the PIs, trial managers and nurses interviewed affiliated to the trials identified from the literature or with different trials? If the latter, what was the purpose of identifying trials? On the other hand, the authors say that 'potential interviewees were identified from the list of authors...' (P 6, L 11)?

See response to point 4 above. The interviewees were identified from the list of authors in the RCTs

identified for our sampling frame or from trial networks.

6. 'All transcripts were reviewed and validated by at least two authors' (P 6, L 45). What do the authors mean by 'validating transcripts'? Was each transcript coded by two authors? If so, what was the degree of convergence/divergence and the procedure used to discuss and resolve differences in coding?

Each transcript was reviewed independently by at least two transcript review group members who documented the emerging major themes which were then discussed by the group in pre-planned meetings. This group met monthly while data were collected. There was a high degree of convergence in the themes identified.

The text now reads as

A transcript review group was formed with four authors (VB, CV, GR, FS) and was heterogeneous in terms of the members' professional background.

Each transcript was reviewed independently by at least two group members who documented the emerging major themes which were then discussed by the group in pre-planned meetings. This group met monthly while data were collected. There was a high degree of convergence in the themes identified.

Data were also analysed iteratively so early results could be incorporated and probed in later interviews to increase the depth of the findings. Labels were agreed and used as broad codes to label textual data associated with major themes. Sub codes were identified discussed and agreed between two authors (FS, VB) and also used to label textual data. Transcripts were subsequently coded by one author (VB). The first two transcripts were checked by another author (FS). A key consideration in coding was to mark data produced spontaneously and that which was specifically asked about, therefore data were coded to take account of the response and the question that prompted that response in order to apply appropriate emphasis on responses. Codes for the six strategies identified by the Cochrane review i.e. "communication", "incentives", "questionnaires", "methodology", "case management" and "behavioural" were used as a priori codes for the later part of the interview transcripts as participants were asked specifically about the use of Cochrane review strategies. Textual data relating to retention strategies was subsequently retrieved from a coded database and constantly compared across the three groups of trial personnel (i.e. PIs TMs RNs) to identify the use of each strategy and to document the reported advantages and disadvantages of each. Any factors thought by the interviewees to contribute to either retention or loss to follow-up were also coded and retrieved for analysis^{12;13}. Relevant quotes, representing the interviewee's views, were selected to illustrate the results. Interviews were conducted between August 2010 and May 2011 until no new content or themes emerged.

7. Again, how were the codes 'verified' by another author? Did the other author actually code transcripts or simply review the codes independent of transcripts?

See response to question 6 above.

8. Details of the mechanics of Atlas use are not needed.

These have been removed

9. I imagine the six strategies were used as a priori codes for the latter part of the interview where the interviewees were asked about these. How were the 'open' sections of the interview coded where

interviewees were asked to talk about any strategies they used? Were the six strategies used as codes for these sections as well?

The six strategies were used as broad a priori codes for the later part of the interview transcripts as participants were asked specifically about the review strategies in that part of the interview. Where interviewees spontaneously mentioned a strategy that they had used in the “open” section of the interview e.g. the use of reminder letters in follow-up, they were probed to talk more about the use of that strategy. This text would have been coded with terms e.g. “spontaneous”, “communication”, and sub codes e.g. “letter”

10. The authors describe a process of constant comparison – if this is what they used, it would be good to indicate this.

Constant comparison was used between the three groups of interviewees and in transcripts within the three groups during analyses.

The text now reads

Textual data relating to retention strategies was subsequently retrieved from a coded database and constantly compared within and across the three groups of trial personnel (i.e. PIs TMs RNs) to identify the use of each strategy and to document the reported advantages and disadvantages of each.

11. A positivistic language is used: ‘advantages and disadvantages’ of each strategy were ‘identified.’ The authors need to acknowledge that they are reporting views and use more appropriate language, i.e. the reported advantages, etc.

This language has been changed

12. I find the analysis superficial and conceptually weak. However, this may reflect the type and function of the study.

The function of the study was to explore the use of different strategies and those identified by our Cochrane review to improve retention in primary care trials. We feel that the approach to the analysis was appropriate to meet the study aim and objectives.

13. Punctuation needs to be reviewed.

The punctuation has been reviewed.

Results:

14. An introductory paragraph with the main findings and the focus of the analysis would be helpful to orient the reader through the many sub-headings.

An introductory paragraph has been added, this reads as

Results are presented first on the interviewee’s experiences of and perspectives on using communication, incentive and new questionnaire strategies, identified by the Cochrane review, to improve primary care trial retention. This is followed by views on other, less frequently evaluated, strategies identified by the Cochrane review i.e. behavioural, case management, and methodological strategies. Factors thought to contribute to retention and loss to follow-up in primary care trials are also presented.

15. I've found it hard to anchor the paper; the problem seems to be that it's trying to do two different things:

a) investigate the use of and views on the Cochrane strategies and

b) explore the everyday experiences and views of trial staff.

From my reading, the most striking finding is the difference between the types of intervention studied in the Cochrane review and the issues that preoccupy staff on the ground and the strategies they have developed. I think a paper structured around this core idea would work much better and the authors need to consider if this is feasible.

Given the general aim of the study (in the Introduction) and the ordering of the questions, with the Cochrane review strategies at the end of the topic guide, I'm not sure why these are presented as the main findings, with the 'open' sections of the interview – interviewees' own experiences and views – left to the end. The authors should consider turning this around, particularly as the open section has produced interesting and important data.

By asking questions in this order the spontaneous reporting of use or views on the methods evaluated in the systematic review was captured before participants were prompted. We feel that given that the aim of our paper is to describe the use and acceptability of using the strategies identified by the Cochrane review, the results relating to those results have been placed 1st in the results section and can be read in conjunction with the Cochrane review itself. The order has therefore remained unchanged.

16. The authors note, in the Introduction, the absence in the Cochrane review of strategies used to encourage trial participants' return to study sites for clinical follow-up (as opposed to strategies used to encourage questionnaire response). They seem to imply that this is part of the rationale for the qualitative study which was designed to address this by including interviewees from primary care trials where this type of retention (return to study sites) was a problem. Yet, they give no information on the nature of the follow-up procedures in the 37 trials. As a result, the reader does not know what type of retention problem (return to site/return of questionnaire) the various strategies discussed by the interviewees were intended to address. Having identified this as an important variable, the authors need to provide information on this.

Different types of follow-up were used in the trials identified for the sampling frame. We have added the different types used in the results section which now reads as

Different types of follow-up were used in these trials: face-to-face follow-up at clinics (n=20), postal questionnaires (n =6), follow-up at home (n=2) or self-completion diaries (n=2). Combinations of follow-up types were also used: clinic visits or postal follow-up (n= 4), telephone and postal follow-up (n=2), register data and follow-up at home (n=1).

17. Similarly, the authors do not report on whether there were any patterns in the data on the basis of level of loss to follow-up (a sampling criterion).

There were few examples of patterns in the data on the basis of level of loss to follow-up apart from for the use of incentives. Such an example has been added to the incentives section which now reads as

TMs and PIs sampled from trials with <20% loss to follow-up use appeared to use more strategies to improve retention than those from trials with ≥20% loss to follow-up. For example those in the <20% group gave incentives at different time points to try to keep the participants motivated to return questionnaires. TMs and PIs sampled from trials with ≥20% loss to follow-up seemed more cautious about the use of incentives and some felt participants could feel coerced if incentives were used to improve follow-up.

18. What the authors have labelled as 'the disease area' (as a factor in retention) may be more appropriately and informatively labelled as 'guilt and shame.'

We agree with this comment. This has been changed and now reads as

Some interviewees thought that if participants did not perceive a benefit from participation they might not return for their follow-up visit. Feelings of guilt and shame with not having achieved positive outcomes in a trial were thought to contribute to loss to follow-up in some disease areas / conditions e.g. in treatment for obesity trials if participants were not losing weight they might not return to be weighed perhaps due to the shame and stigma associated with not achieving their expected aims. Retention in trials involving behaviour change was also thought to be problematic if change targets were not achieved by participants. For example, interviewees were generally in agreement that retention can be problematic in trials involving participants with a mental health issue.

Box 1 and Box 2, as currently organised, are confusing and they fragment the reader's attention as the main text already has these themes and relevant data extracts. If data extracts for the themes are to be put in a box, then they should all be put in a Box. It's confusing to have extracts in the main text and then more in a Box.

We have added all the quotes to the text and removed boxes 1 and 2

Tables 3 & 4 don't work either.

Some information is repeated in the main text; the information includes: who used the strategy, whether it was mentioned spontaneously, how it was used and views on effectiveness, but in a haphazard way, i.e. not all these domains are covered in each cell. The result is a confusing collection of disparate information.

Table 4, in particular, reads like (under-analysed) descriptive summaries of views and experiences on different aspects, reported in different ways. Tables could be used to supplement (not repeat) the main text and need to provide information in a consistent manner.

We have removed tables 3 and 4 and reorganised these sections in the text

Minor points:

P7 L25: Should '20' be '17'

We were unclear about which section the reviewer was referring to here.

P11 L44-48. What does this sentence mean – which aspect of the questionnaire could influence the response?

We were unclear about which section the reviewer was referring to here.

P14 L43 – there seems to be text missing here.

We were unclear about which section the reviewer was referring to here.

Referring to the Cochrane systematic review as the Cochrane review may make the source clearer in each instance

We have referred to the Cochrane review throughout the manuscript

Table 3: Should the distinction between 'pharmaceutical' and 'academic' trials be: 'industry sponsored' and 'publicly funded'?

Table 3 has been removed

Table 4: 'Speculation'? Or 'experiences and views'?

Table 4 has been removed

Discussion:

19. The authors need to emphasize the fact that only two (incentives and communication) of the six Cochrane review strategies were mentioned spontaneously. On the other hand, several other factors and ways of improving retention were identified that were not covered in the Cochrane review. Some, e.g. the relationship/communication between trial staff and trial participants, are clearly more difficult to design interventions for and to evaluate. It seems clear that trial staff oriented to patient concerns and needs (as they perceived them) and had a high level of empathy. This seems an important finding with implications for future intervention design as well as the appropriateness of existing ones (behavioural strategies), but is not picked up in the Discussion.

These points have been added to the discussion

The Discussion has to be revised if the authors choose to reorganise and refocus the paper as suggested here.

Reviewer Name Margaret Cupples

Institution and Country Queen's University, Belfast

Please state any competing interests or state 'None declared': None declared

Abstract and Introduction

20. The research question and aim are clearly defined at the end of the introduction but could be better clarified in the abstract.

We have made our aim clearer in the abstract.

Methods

21. In respect of the methods, there is inconsistency in description of 'trials' and 'randomised trials' - this should be clarified - that all studies were RCTs?

We have clarified that all studies were RCTs in the text

22. It should also be noted that all nations in the UK were not included- studies selected were

confined to England and Scotland.

Published RCTs from Northern Ireland and Wales were not identified through the sources searched although sites from these countries may have been included in the trials identified for the sampling frame. Furthermore interviewees may nevertheless have worked on trials conducted in Northern Ireland and Wales. Their experiences are drawn from trials they had worked on generally.

23. Also, no information is given about those who were selected for interview but declined to participate - this should be included.

In the results section further information has been added about those who declined to participate and non-responders.

This now reads

Fifty four trial personnel were identified for recruitment from 37 primary care trials from England and Scotland that were included in the sampling frame. Eleven of the fifty four invitees declined to participate (PIs (n=10), TMs (n=1)). Seven of these PIs recommended another co-author to invite. Fourteen of the fifty four did not respond (PIs (n=10), TMs (n=3), RNs (n=1)). Ten PIs, ten TMs and nine RNs were recruited from 23 of the 37 primary care trials identified.

24. The description of identification of participants would be better if attention was paid to detail a logical flow of the process followed was given - perhaps a table could show the sources from which participants recruited -and the relevant trials?

We chose not to include a table with the RCTs from which the researchers were recruited as this could potentially lead to interviewees being identified. We have made the process we used to identify potential interviewees clearer see response to number 4.

25. Also - how were trials with 20% loss classified - or were they excluded?

One RCT included in the sampling frame did have a 20% response rate. This was included in the $\geq 20\%$ response rate. This has been change in the methods section under sampling the study population to read as

The trial publications identified spanned different disease areas, research units, and levels of loss to follow-up (Tables 1 and 2) and were stratified by levels of loss to follow-up ($<20\%$, $\geq 20\%$), and publication date (2000-2004 and 2005-2010).

And what about the publications from 2005?

RCTs from 2005 were included in the sampling frame. We have removed pre and post 2005 and added the categories for the dates of publication used i.e. 2000-2004 and 2005-2010, see response to number 25 above.

26. How was consent obtained?

Participants were given an information sheet with the letter of invitation. If they agreed to a face to face interview before the interview commenced the potential interviewee was given a copy of the information sheet to read and a short explanation of the purpose of the research and their role in it. They were then asked to sign a consent form and given a copy for their own records. Original consent forms are filed at the MRC coordinating centre. This has been clarified in the data collection section

27. The setting of interviews (were they face-to face?) may be added to the section on data collection.

We have added the setting to the data collection section of the methods section

Results

I have said that the results are not presented clearly - because I feel they could be presented much more clearly.

28. Does the statement that interviewees were from the 'fields of' - does this mean they had specialist interests in specific areas - or does this refer to the publication from which they were identified, or the studies which they used in their responses?

This refers to the RCT publication from which the interviewee was identified. We have clarified this in the results section

29. Perhaps a version of Tables 3 and 4 could be included and referred to early in the results, to help to clarify.

Tables 3 and 4 have been removed, and are now illustrated in the text

30. I found it was confusing to read about types of incentives/ values/ ethics committees' views/ perceptions of use/timing/alternatives/ethics approval - all in one section. More time taken to consider how these findings could be considered to report the 'use of strategies' and them to report 'factors thought to contribute to retention' - i.e. a clear statement of findings regarding the perceived value of incentives would be helpful.

Trial personnel were uncertain of the effectiveness of the incentives used, and we have therefore added a statement about this to the findings at the beginning of the incentives section. The incentives section has also been re organised in light of the reviewers comments.

31. With regard to Tables 3 and 4, it is not clear why they are divided into the 3 groups- nor where the different groups share experiences / views re strategies. Consideration of a more organised framework for reporting these would be helpful, as would a consistent approach to the detail reported regarding responses.

We have removed tables 3 and 4 as there was considerable overlap between these and the text

32. Similar comments re organisation of the report of findings applies to each of the sections - communication/ questionnaire format/ - to make the actual findings more easily identified within the text.

We have rearranged the findings in these sections

33. In respect of the section re the use of either strategies - I think these needs more detail - I find it difficult to understand how 'Arranging workshops about goal-setting.....' relates to a retention strategy rather than being a component of the intervention being tested??

We have added more detail to help explain these strategies in the other strategies section

Discussion

34. The opening paragraph of the discussion suggests that age/ condition/ research environment

contribute to loss of follow-up - but the specific characteristics of these variables which are relevant need to be identified in order to provide meaningful information.

We have added specific characteristics of these variables in the opening paragraph of the discussion.

35. In respect of the discussion of limitations, again I'm not sure why there's value in a study which focuses only on retention rather than recruitment as well? Can this be elucidated?

Whilst we appreciate that recruitment is an important methodological research area for trial conduct, we felt that this was outside the scope of our project which was to focus specifically on retention in primary care randomised trials which has received little attention in the literature to date. We feel it is an important area for consideration because loss to follow up can lead to incomplete ascertainment of the primary outcome which can bias results.

We have added the following sentence to the discussion

Trial retention is important because loss to follow up can lead to incomplete ascertainment of the primary outcome, bias results and impact on the generalizability of trial findings. There has been little research to date on the acceptability and use of retention strategies.

36. Or perhaps be explicit regarding how this paper adds value to previous work in the UK and how these findings compare with previous reports?

Please refer to our response to point 35

37. The use of the wording 'RCTs in general' may be reviewed, to be more specific about what 'in general' means?

Here we mean research in other areas outside of primary care. We have modified this in the discussion.

38. The section headed 'meaning and implications' would be more clearly presented with sub-headings and this may help improve the coherence within sub-sections. (There is a typo - line 4 of page 18 - 'if' - should be 'of'.)

We have made the meaning and implications heading clearer.

39. I find the section re integrating qualitative research with systematic reviews rather vague - this could be much more meaningful if specific examples were used to illustrate statements.

We have added examples to illustrate the statements made in this section

40. Lastly, the focus in the conclusion is on incentives and 1st class post but the previous section highlights the use of text and email and I found comments regarding reminders, harassment, engagement, rapport, patient benefit and respect for patients' time all very useful pointers towards improving retention - could these be integrated in the conclusion?

We have integrated these other factors into the conclusion

I am aware that I am being critical of this paper - but I do feel that it has value which could be much improved by added attention to detail and organisation of its presentation.

This is an interesting study but I feel its contribution to the literature would be much improved by some further work. The introduction itself does not read easily - small changes to the use of English would help its fluency and allow better coherence in the development of the rationale for the study. The closing words - 'trials in general' is a rather loose statement and should be better defined. A minor point re grammar - traditionally the correct use of data is plural - thus data 'were' rather than data 'are' would be correct.

We have revised the whole paper with the aim of making it clearer to the reader.

VERSION 2 – REVIEW

REVIEWER	Cupples, Margaret Queens University, Belfast, General Practice
REVIEW RETURNED	17-Nov-2013

GENERAL COMMENTS	The problem with the definition of the study objective lies in the inclusion (in abstract, and on p5, line 27)/exclusion (page 6, line1) of the word 'randomised' - the methods appears to indicate that all types of trial conducted in primary care were included? I feel the abstract 'Results' section requires a little further revision. The sentence 'Shorter questionnaires were used, and suggestions offered for improving questionnaire design.' needs elaboration in order to inform the reader about useful relevant findings - was there a guide re 'shortness'; what were the suggestions? Also see lines 28/29 - was it being teenage that contributed to poor retention?/ what disease type/ what aspects of work environment were relevant? Adding such detail (or revising this section) would render these statements more helpful to researchers wishing to decide if this paper is worth reading in order to inform their future practice. On page 4, the Conclusion of the abstract, reference is made to 'the Cochrane review' - this is not meaningful - no earlier reference to any review now exists. Which communication strategies need re-consideration? - this conclusion does not appear to be based within the abstract's results section. (Note also typos on line 22 of page 3 and line 3 of page 4) Methods - Note typo p7, line 16 - 'that...was' - correct references to the word 'data' should be to plural - thus better to write 'those' and 'were'. This comment also applies to line 23. Re results - p8, line 5 -would it be more correct to write that 'interviewees had participated in trials conducted in.....and amend line 7 accordingly? Can 'register data' (line 15) be explained- does this refer to registered post? Mention of the Cochrane review (line 29/31) needs reference. Please consider lay-out of headings/subheadings to aid clarity - see below. I'd suggest retaining 'Other' in heading 'Factors thought to contribute to trial retention' - which seems to fit in sequence as sub-heading Number 5? Page 16, lines 20-32 relate to communication strategies - could/ should these not be included in the earlier relevant section? Page 18 - on first reading, the heading 'Factors thought to contribute to loss to follow-up' seemed to lose the focus on retention (consistent use of terminology is good) and appeared to 'fit' as sub-heading Number 6? But perhaps this is just another sub-theme
---

within the 'Factors thought to contribute to trial retention'? Perhaps consideration could be given to a heading such as 'Participants' personal characteristics' - and include altruism appropriately within this? Overall this section could be written more succinctly without loss of important detail. Line 44 refers to benefit from participation - this may belong within the earlier relevant section?
Page 19 - the 'working environment' (line 34) - appears to be a separate issue to those immediately preceding it and may be given a separate sub-heading?

Discussion -Page 20, lines 32/33 - I'd consider that a summary statement is more appropriate as a principal finding - it really is personal characteristics (such as interest/altruism/mobility/ time constraints) that encourage retention. 'the age of participants e.g. young men and teenagers; the disease area / condition e.g. mental illness and behavioural problems' is not a useful generalisation/summary, in my opinion.

The statement on Page 20 line 42 and p 21 line 1, that this study is 'the first to explore the use of retention strategies in UK primary care trials and the factors associated with retention' is not quite true - see Leatham CS et al. ('Identifying strategies to maximise recruitment and retention of practices and patients in a multicentre randomised controlled trial of an intervention to optimise secondary prevention for coronary heart disease in primary care. BMC Medical Research Methodology 2009, 9:40.doi.10.1186/1471-2288-9-40'). I think this statement could be modified, perhaps to say that it focuses on retention, rather than recruitment strategies - or omit the sentence without any loss of impact: the revised discussion, and preceding sentence, emphasising the importance of retention is good.

Key messages -Page 25, lines 23/24 - as mentioned above I do not consider that this : ('participant age, disease/condition and research environment') provides useful information - better to make a definitive statement such as in your conclusion - or report findings re lack of interest, time constraints, physical constraints and limitations of research facilities.

Also p25, again I would suggest that this is not the first study to explore the use of retention strategies in UK primary care - though it is the largest study to date to do so. Rather, I think its strength is that it adds to previous knowledge regarding the use of retention strategies within research trials in UK primary care.

Note typo page 14, line 15 - behaviour.

VERSION 2 – AUTHOR RESPONSE

Reviewer Name Margaret Cupples

Institution and Country Queen's University, Belfast Please state any competing interests or state

'None declared': None declared

The problem with the definition of the study objective lies in the inclusion (in abstract, and on p5, line 27)/exclusion (page 6, line1) of the word 'randomised' - the methods appears to indicate that all types of trial conducted in primary care were included?

Randomised trials were included in the sampling frame, trialists interviewed were sampled from randomised trials included in the sampling frame. We have added randomised to page 6 line 1.

I feel the abstract 'Results' section requires a little further revision. The sentence 'Shorter questionnaires were used, and suggestions offered for improving questionnaire design.' needs elaboration in order to inform the reader about useful relevant findings - was there a guide re 'shortness'; what were the suggestions? Also see lines 28/29 - was it being teenage that contributed to poor retention?/ what disease type/ what aspects of work environment were relevant? Adding such detail (or revising this section) would render these statements more helpful to researchers wishing to decide if this paper is worth reading in order to inform their future practice. On page 4, the Conclusion of the abstract, reference is made to 'the Cochrane review' - this is not meaningful - no earlier reference to any review now exists. Which communication strategies need re-consideration? - this conclusion does not appear to be based within the abstract's results section. (Note also typos on line 22 of page 3 and line 3 of page 4)

The abstract has been revised

Methods - Note typo p7, line 16 - 'that...was' - correct references to the word 'data' should be to plural - thus better to write 'those' and 'were'. This comment also applies to line 23.

These have been changed

Re results - p8, line 5 -would it be more correct to write that 'interviewees had participated in trials conducted in.....and amend line 7 accordingly?

This has been changed and now reads:

Interviewees involved in running primary care trials conducted in nutrition, health promotion, neurology, gynaecology, mental health, musculoskeletal, ear nose and throat (ENT), respiratory, endocrine medicine and trials of minor medical conditions.

Can 'register data' (line 15) be explained- does this refer to registered post?

This refers to "registry data" and has been changed

Mention of the Cochrane review (line 29/31) needs reference.

The reference has been added

Please consider lay-out of headings/subheadings to aid clarity - see below. I'd suggest retaining 'Other' in heading 'Factors thought to contribute to trial retention' - which seems to fit in sequence as sub-heading Number 5? Page 16, lines 20-32 relate to communication strategies - could/ should these not be included in the earlier relevant section? Page 18 - on first reading, the heading 'Factors thought to contribute to loss to follow-up' seemed to lose the focus on retention (consistent use of terminology is good) and appeared to 'fit' as sub-heading Number 6? But perhaps this is just another

sub-theme within the 'Factors thought to contribute to trial retention'?

Perhaps consideration could be given to a heading such as 'Participants' personal characteristics' - and include altruism appropriately within this? Overall this section could be written more succinctly without loss of important detail.

This section has been revised

Line 44 refers to benefit from participation - this may belong within the earlier relevant section?

This has been removed

Page 19 - the 'working environment' (line 34) - appears to be a separate issue to those immediately preceding it and may be given a separate sub-heading?

The 'working environment' has been added as a sub heading

Discussion -Page 20, lines 32/33 - I'd consider that a summary statement is more appropriate as a principal finding - it really is personal characteristics (such as interest/altruism/mobility/ time constraints) that encourage retention. 'the age of participants e.g. young men and teenagers; the disease area / condition e.g. mental illness and behavioural problems' is not a useful generalisation/summary, in my opinion.

This now reads

Personal characteristics e.g. interest in the trial, altruism and time, encourage retention

The statement on Page 20 line 42 and p 21 line 1, that this study is 'the first to explore the use of retention strategies in UK primary care trials and the factors associated with retention' is not quite true - see Leathem CS et al. ('Identifying strategies to maximise recruitment and retention of practices and patients in a multicentre randomised controlled trial of an intervention to optimise secondary prevention for coronary heart disease in primary care. BMC Medical Research Methodology 2009, 9:40.doi.10.1186/1471-2288-9-40'). I think this statement could be modified, perhaps to say that it focuses on retention, rather than recruitment strategies - or omit the sentence without any loss of impact: the revised discussion, and preceding sentence, emphasising the importance of retention is good.

Leathem's very informative paper described the use of retention and recruitment strategies in one primary care retention trial. Our study is the first to explore the use of the spectrum of strategies to improve retention across randomised primary care trials.

Key messages -Page 25, lines 23/24 - as mentioned above I do not consider that this : ('participant age, disease/condition and research environment') provides useful information - better to make a definitive statement such as in your conclusion - or report findings re lack of interest, time constraints, physical constraints and limitations of research facilities.

This now reads:

Lack of time and interest in the trial, physical constraints, and limitations of research facilities

Also p25, again I would suggest that this is not the first study to explore the use of retention strategies in UK primary care - though it is the largest study to date to do so. Rather, I think its strength is that it

adds to previous knowledge regarding the use of retention strategies within research trials in UK primary care.

Note typo page 14, line 15 - behaviour.

This has been changed